# Description and Genome-Based Analysis of *Vibrio chaetopteri* sp. nov., a New Species of the Mediterranei Clade Isolated from a Marine Polychaete

**DOI:** 10.3390/microorganisms13030638

**Published:** 2025-03-11

**Authors:** Valeriya Kurilenko, Evgenia Bystritskaya, Nadezhda Otstavnykh, Peter Velansky, Darina Lichmanuk, Yulia Savicheva, Lyudmila Romanenko, Marina Isaeva

**Affiliations:** 1G.B. Elyakov Pacific Institute of Bioorganic Chemistry, Far Eastern Branch, Russian Academy of Sciences, Prospect 100 Let Vladivostoku, 159, Vladivostok 690022, Russia; ep.bystritskaya@yandex.ru (E.B.); chernysheva.nadezhda@gmail.com (N.O.); lichmanyukdarina@gmail.com (D.L.); iu.savicheva0@yandex.ru (Y.S.); lro@piboc.dvo.ru (L.R.); 2A.V. Zhirmunsky National Scientific Center of Marine Biology, Far Eastern Branch, Russian Academy of Sciences, Palchevskogo Street 17, Vladivostok 690041, Russia; velansky.pv@gmail.com

**Keywords:** marine polychaete, Mediterranei clade, *Vibrio chaetopteri*, Troitsa Bay, genome

## Abstract

Two novel strains, CB1-14^T^ and CB2-10, were isolated from the marine polychaetes *Chaetopterus cautus* from the Sea of Japan. Phylogenetic analysis based on the 16S rRNA sequences revealed that the two strains belong to the genus *Vibrio*, sharing 98.96% identity with *Vibrio hangzhouensis* CN 83^T^. MLSA using five protein-coding genes (*ftsZ, gyrA, gyrB, mreB*, and *rpoA*) showed that CB1-14^T^ and CB2-10 are closely related to the members of the Mediterranei clade, namely *Vibrio mediterranei* CECT 621^T^, *Vibrio barjaei* 3062^T^, *Vibrio thalassae* CECT 8203^T^, *Vibrio hangzhouensis* CGMCC 1.7062^T^, *Vibrio maritimus* CAIM 1455^T^, and *Vibrio variabilis* CAIM 1454^T^. Based on both MLST neighbor-net phylogenetic network and phylogenomic tree results, they fell into the subclade formed by *V. maritimus* CAIM 1455^T^ and *V. variabilis* CAIM 1454^T^. Both new strains CB1-14^T^ and CB2-10 showed the highest ANI/AAI values of 91.3%/92.7% with *V. variabilis* CAIM 1454^T^ and 90.3%/93.1% with *V. maritimus* CAIM 1455^T^. The dDDH values between strain CB1-14^T^ and the members of the Mediterranei clade ranged from 20.9% to 45.7%. Major fatty acids were C_16:1_*ω*9*c*, C_16:1_*ω*7*c*, and C_18:1_*ω*9*c*, followed by C_16:0_ and C_18:1_*ω*7*c*. The genome of CB1-14^T^ is 5,591,686 bp in size, with DNA G+C content of 46.1%. It consists of two circular chromosomes (3,497,892 and 1,804,652 bp) and one plasmid (241,015 bp) and comprises 4782 protein-coding genes and 10 *rrn* operons. The CB1-14^T^ and CB2-10 genomes were enriched in CAZyme-encoding genes of the following families: GH1, GH3, GH13, GH23, GH43, GH94, PL17, and CE4, indicating the potential to catabolize alginate, xylan, and chitin, common polysaccharides in marine ecosystems. Based on the combined phylogenomic analyses and phenotypic properties, a new species, *Vibrio chaetopteri* sp. nov., is proposed, with CB1-14^T^ = (KMM 8419^T^ = KCTC 92790^T^) as the type strain.

## 1. Introduction

The genus *Vibrio* was proposed by Pacini in 1854, with *Vibrio cholerae* as the type species, and it currently includes 153 validly published species and 23 species with “preferred name” taxonomic status (https://lpsn.dsmz.de/genus/vibrio, accessed on 16 December 2024) [1]). *Vibrio* are ubiquitous in fresh aquatic, marine, brackish environments, in association with humans and animals, diverse and highly heterogeneous in their metabolic properties [2,3]. A recent update of the *Vibrionaceae* family phylogenetics using 191 bacterial genomes delineated it into 51 distinct clades including 21 newly defined ones [4]. However, it has been shown that multilocus sequence analysis (MLSA) based on eight house-keeping genes [5] can still be reliably used to classify new species and strains [4]. The Mediterranei clade was firstly identified as a monophyletic clade consisting of three species, *V. mediterranei*, *V. maritimus*, and *V. variabilis* [6]. Later, the clade was expanded by two species, *V. thalassae* and *V. barjaei*, with subsequent division into two subclades based on pan-genome analysis [7]. According to MLSA [4], the Mediterranei clade is represented by two major branches: subclade 1 formed by *V. thalassae*, *V. mediterranei*, and *V. barjaei* and subclade 2 formed by *V. variabilis*, *V. maritimus*, and *V. hangzhouensis*.

The species *V. mediterranei* was originally isolated from various marine samples such as sea sediment, seawater, and plankton in the coastal area of Valencia, Spain [8]. It is now recognized as a species with a worldwide distribution in marine environments, as it has been isolated from seawater and sediments, different bivalves (mussels, clams, oysters, and ark shells), other marine invertebrates (corals, fireworms, sponges, shrimps, and sea urchins), and fish (turbot, amberjack, and spotted rose snapper) [9]. *V. barjaei* was isolated from different culture stages of carpet shell clam (*Ruditapes decussatus*) reared in a bivalve hatchery (Galicia, NW Spain) [10]. *V. thalassae* was isolated from different samples of coastal seawater at Malvarrosa Beach, Valencia and Vinaroz, Castellón, Spain [9]. *V. hangzhouensis* was found in a marine sediment sample from Zhejiang, China [11]. *V. maritimus* CAIM 1455^T^ and *V. variabilis* CAIM 1454^T^ were isolated from mucus of the zoanthid *Palythoa caribaeorum* in Portinho Beach and in Preta Beach, respectively, São Sebastião Channel, São Paulo, Brazil [12].

In the course of our studies, several marine bacteria *Vibrio* spp. isolated from a mucus sample from the food net of the marine polychaetes *Chaetopterus cautus* (formerly identified as *Chaetopterus variopedatus*) collected in Troitsa Bay, Peter the Great Gulf, Sea of Japan [13], have been analyzed.

In this study, two new bacterial strains of *Vibrio*, designated as strains CB1-14^T^ and CB2-10, were characterized using a polyphasic approach. Phylogenomic and pan-genome analyses were performed to gain insight into the phylogenetic relationships and metabolic potential of strains CB1-14^T^ and CB2-10. Based on the combined phylogenomic analyses and phenotypic properties, a new species, *Vibrio chaetopteri* sp. nov., is described.

## 2. Materials and Methods

### 2.1. Isolation and Phenotypic Characterization of Bacteria

The specimens of marine polychaetes *Chaetopterus cautus* were dug out of sea sand at a depth of 6–10 m by scuba divers (salinity 33%, temperature 20 °C) in Troitsa Bay, Peter the Great Gulf, Sea of Japan, Russia (42°37.380597′ N, 131°7.606052′ E) in August 2016. Strains CB1-14^T^ and CB2-10 were isolated as described previously [13] and stored at −70 °C in Marine Broth 2216 (MB; BD Difco^TM^, Sparks, MD, USA) supplemented with 20% (*v*/*v*) glycerol. Strains CB1-14^T^ and CB2-10 have been deposited to the Collection of Marine Microorganisms (KMM), G. B. Elyakov Pacific Institute of Bioorganic Chemistry, Far Eastern Branch of the Russian Academy of Sciences, Vladivostok, Russia, as KMM 8419^T^ and KMM 8420, respectively, and strain KMM 8419^T^ was deposited as the type strain to the Korean Collection for Type Cultures (KCTC), Korea Research Institute of Bioscience and Biotechnology, Republic of Korea, under number KCTC 92720^T^. The type strain *V. thalassae* KCTC 32373^T^ was kindly provided by the Korean Collection for Type Cultures, Republic of Korea. All strains used in this study for phenotypic tests and lipid analyses were grown on/in MB 2216 and Marine Agar 2216 (MA; BD Difco^TM^, Sparks, MD, USA), if not stated otherwise. Gram-staining, oxidase, catalase, and motility tests were examined by the hanging drop method, according to the standard methods described in [14].The morphology of cells negatively stained with a 1% phosphotungstic acid was examined by the electronic transmission microscope Libra 120 FE (Carl Zeiss, Oberkochen, Germany), provided by the A.V. Zhirmunsky National Scientific Center of Marine Biology, Far Eastern Branch, Russian Academy of Sciences, using cells grown in MB 2216 on carbon-coated 200-mesh copper grids. The tests, including hydrolysis of starch, gelatin, L-tyrosine, chitin, and nitrate reduction (sulfanilic acid/α-naphthylamine test), with growth at different salinities (0–10% NaCl), temperatures (5–40 °C), and pH values (4.0–12.0), were carried out as described in [14]. The medium MA 2216 (or MB 2216) was used as a basal for determination of substrate hydrolysis, temperatures, and pH. Hydrolysis of starch, casein, and Tweens 20, 40, and 80 was tested as described in a previous paper [15]. Hydrolysis of DNA was examined using DNase Test Agar (BD BBL^TM^, Sparks, MD, USA). Formation of H_2_S from thiosulfate was studied using a lead acetate paper strip. Biochemical tests for all strains studied using the API 20E, API 20NE, and API ZYM test (bioMérieux, Marcy-l’Étoile, France) were performed as described by the manufacturer.

Antibiotic susceptibility of strains studied was examined on MA 2216 plates using commercial paper discs (Research Centre of Pharmacotherapy, St. Petersburg, Russia) impregnated with the following antibiotics (μg per disc, unless otherwise indicated): ampicillin (10), benzylpenicillin (10 U), vancomycin (30), gentamicin (10), kanamycin (30), carbenicillin (100), chloramphenicol (30), neomycin (30), oxacillin (10), oleandomycin (15), lincomycin (15), ofloxacin (5), rifampicin (5), polymyxin (300 U), streptomycin (300), cephazolin (30), cephalexin (30), erythromycin (15), nalidixic acid (30), tetracycline (30), and doxocycline (30). A suspension of cells of the studied strains at a concentration of 0.5 McFarland was applied to a Petri dish with MA 2216 medium, and then discs with antibiotics were placed and incubated at a temperature of 25°C for 48 h.

For polar lipid and fatty acid analyses, strain KMM 8419^T^, KMM 8420, and *V. thalassae* KCTC 32373^T^ were cultivated on MA 2216 at 24 °C for 48 h. Lipids were extracted using the extraction method [16]. Two-dimensional thin-layer chromatography of polar lipids was carried out on silica gel 60 F 254 (10 × 10 cm, Merck, Darmstadt, Germany) using chloroform–methanol–water (65:25:4, *v*/*v*) for the first direction, and chloroform–methanol–acetic acid–water (80:12:15:4, *v*/*v*) for the second one [17]. Lipids were detected by sequentially spraying with 0.25% ninhydrin in acetone (for detecting amino-group-containing lipids), molybdate reagent (for detecting phospholipids), and 5% sulphuric acid in methanol, followed by heating at 130 °C. Before TLC, the extracts were redissolved in methanol:chloroform (9:1) to precipitate the interfering non-polar compound present in large quantities in the samples. Respiratory lipoquinones were analyzed by the reversed-phase HPLC using the modified method [18]. A Shimadzu LC–30 chromatograph with a photodiode array detector (SPD-M30A), equipped with Shimpack ODS II (150 × 2.1 mm) column, was used. The column temperature was 40 °C, and isocratic elution with methanol–isopropanol (7:3) was used. Fatty acid methyl esters (FAMEs) were prepared according to the procedure of the Microbial Identification System (MIDI) [19]. The analysis of FAMEs was performed using the GC-2010 chromatograph (Shimadzu, Kyoto, Japan) equipped with a capillary column SH-Rtx-5ms (30 m × 0.25 mm I.D.) (Shimadzu, Kyoto, Japan), and the temperature was programmed from 160 °C to 250 °C, at a rate of 2 °C/min. Identification of FAMEs was accomplished by equivalent chain length values and comparing the retention times of the samples to those of standards. In addition, FAMEs were analyzed using a GC–MS Shimadzu model QP2020 (with column SH-Rtx-5ms and the temperature program from 160 °C to 250 °C at a rate of 2 °C/min).

### 2.2. 16S rRNA Gene Sequence Analysis

The 16S rRNA gene of strains CB1-14^T^ (=KMM 8419^T^) and CB2-10 (=KMM 8420) was amplified with primer pair 27F (5′-AGAGTTTGATCMTGGCTCAG-3′) and 1492R (5′-TACGGTTACCTTGTTACGACTT-3′) [20] and sequenced on an ABI 3130xl Genetic Analyzer (Applied Biosystems, Hitachi, Tokyo, Japan) as described previously [13]. The similarities of 16S rRNA gene sequences between strains were calculated on the EzBioCloud server [21], and the phylogenetic relationships were estimated on the GGDC web server (http://ggdc.dsmz.de/, updated on 13 December 2024) [22] using the DSMZ single-gene pipeline [23]. Maximum likelihood (ML) and maximum parsimony (MP) trees were inferred from the alignment with RAxML [24] and TNT [25], respectively, with bootstrap analysis of 1000 replicates.

### 2.3. Multilocus Sequence Analysis (MLSA)

To select KMM-8419^T^-related strains, the PCR primers for MLSA (Appendix A) were developed on the basis of gene sequences of FtsZ (cell division protein), GyrA and GyrB (DNA gyrase, subunits A and B), MreB (cell-shape-determining protein), and RpoA (DNA-directed RNA polymerase, subunit alpha) retrieved from the KMM 8419^T^ genome sequence using the Vector NTI version 11.0 (Invitrogen, Carlsbad, CA, USA). Sequences of the KMM 8420 MLST loci were amplified and sequenced with the gene-specific primers using SeqStudio™ Genetic Analyzer (Thermo Fisher Scientific, Waltham, MA, USA). The gene sequences of other type strains of the genus *Vibrio* were taken from their genomic sequences deposited in NCBI datasets. The concatenated sequences based on the five loci were used to conduct phylogenetic analysis using MEGA X software, version 10.2.1 [26]. The genetic distances for MLSA analyses were measured using the Kimura 2-parameter model with 1st + 2nd + 3rd + non-coding codon positions [27]. Split decomposition analysis was carried out using SplitsTree version 4.14.6 [28] as previously described [13].

### 2.4. Whole-Genome Sequencing and Genome Characterization

Genomic DNA was extracted from strains CB1-14^T^ (=KMM 8419^T^) and CB2-10 (=KMM 8420) using the NucleoSpin Tissue kit (Macherey–Nagel, Düren, Germany). The DNA quality was estimated by agarose gel electrophoresis, and the DNA quantity was measured on the Qubit 4.0 Fluorometer (Thermo Fisher Scientific, Singapore, Singapore). Whole-genome sequencing, assembly, and annotation of KMM 8419^T^ were previously performed as described in [29]. The DNA library for KMM 8420 was prepared with Nextera DNA Flex kits (Illumina, San Diego, CA, USA) and sequenced on an Illumina MiSeq instrument using paired-end runs with a 150 bp read length. The nanopore library was prepared for KMM 8420 using EXP-NBD104 and SQK-LSK109 kits (Oxford Nanopore Technologies, Oxford, UK) and sequenced on MinION (Oxford Nanopore Technologies, Oxford, UK). The reads were trimmed using Trimmomatic version 0.39 [30] and their quality assessed using FastQC version 0.11.8 (https://www.bioinformatics.babraham.ac.uk/projects/fastqc/, accessed on 8 February 2024). Hybrid assembly of the KMM 8420 genome was performed using Unicycler v0.4.8 [31] with default parameters. Sequencing depth was estimated using SAMtools version 1.3 [32]. The genome completeness and contamination were accessed with CheckM version 1.1.3 [33]. Gene annotation was performed using various pipelines and systems (NCBI PGAP [34], Prokka [35], RAST [36], EggNOG-Mapper [37]).

The phylogenetic analysis was performed with PhyloPhlAn version 3.0.1 using 400 conserved protein sequences, and an ML tree was reconstructed by RAxML version 8.2.12 under the LG + Γ model with non-parametric bootstrapping of 100 replicates [24,38]. The pan-genome and metabolism analyses of the Mediterranei clade strains were carried out using the microbial pangenomics workflow in Anvi’o version 8 (minbit = 0.5; mcl-inflation = 2; min-occurrence = 1) as described at https://merenlab.org/2016/11/08/pangenomics-v2/, accessed on 20 November 2024 [39]. To calculate the average nucleotide identity (ANIm) between strains, the program ‘anvi-compute-genome-similarity’ with ‘--program pyANI’ flag was used. The amino acid identity (AAI) and in silico DNA–DNA hybridization (dDDH) values between strains were obtained on the online server ANI/AAI-Matrix [40] and TYGS platform [41], respectively.

To predict carbohydrate-active enzymes (CAZymes), dbCAN3 meta server version 10 with default settings was used (http://cys.bios.niu.edu/dbCAN2, accessed on 11 November 2024) [42]. CAZyme genes were selected for further analysis if they were predicted by at least two algorithms. CAZyme-containing gene clusters (CGCs) and polysaccharide-utilizing loci (PULs) were annotated on the dbCAN-PUL meta server [43]. Annotation of secondary metabolite biosynthetic gene clusters was performed on the antiSMASH server, version 7.0 (https://antismash.secondarymetabolites.org/#!/start, accessed on 10 December 2024) [44]. Identification of the secretion system components was conducted with MacSyFinder version 2.1.4 (TXSScan-1.1.3) [45]. The heatmaps and bar plots were visualized using the pheatmap version 1.0.12 and ggplot2 version 3.5.1 packages in RStudio version RStudio/2024.09.1+394 with R version 4.4.2. Fonts and sizes in all figures were edited manually in Adobe Photoshop CC 2018 for better visualization. The circular genomes of strains KMM 8419^T^ and KMM 8420 were visualized using the Proksee platform [46].

## 3. Results and Discussion

### 3.1. Phylogenetic Analyses

The 16S rRNA gene sequences of strains CB1-14^T^ (=KMM 8419^T^) and CB2-10 (=KMM 8420) submitted to the GenBank under accession numbers MK598731 and MK598715 [13] were more than 99% identical to 10 and 11 copies of the 16S rRNA gene retracted from their genomic sequences, respectively. The 16S rRNA phylogenetic analysis showed that both strains clustered together and were closely related to *V. hangzhouensis* CN 83^T^ (98.96% identity) (Figure 1A). The topological structure of the phylogenetic network based on concatenated multilocus sequences using five house-keeping genes (Figure 1B) did not correspond to the structure of the 16S rRNA tree (Figure 1A) and indicated that the new strains CB1-14^T^ and CB2-10 could belong to the Mediterranei clade. Pairwise distances between these new strains and other clade members were 0.042 with *V. variabilis* CAIM 1454^T^, 0.058 with *V. maritimus* CAIM 1455^T^, 0.083 with *V. hangzhouensis* CGMCC 1.7062^T^, 0.095 with *V. mediterranei* CECT 621^T^, and 0.096 with *V. barjaei* 3062^T^ (Appendix A). To confirm the relatedness of the new strains to the Mediterranei clade, eight-gene MLST based on 35 *Vibrio* clade representatives was additionally performed according to Jiang et al. (2022) [4]. The resulting MLST phylogenetic network showed clustering of CB1-14^T^ and CB2-10 with *V. mediterranei* NBRC 15635^T^ (Appendix A), confirming the above MLST results for the five genes.

Based on the results of the 16S rRNA similarity analysis and MLSA, new strains CB1-14^T^ and CB2-10 might belong to a new species of the genus *Vibrio*.

The phylogenomic tree based on 400 concatenated sequences extracted from the genomes of the closely related type strains of the genus *Vibrio* showed that strains CB1-14^T^ and CB2-10 formed a distinct line within the Mediterranei clade (Figure 2A). More precisely, they fell into the subclade formed by *V. maritimus* and *V. variabilis*. In contrast to MLSA results [4], the ML genomic tree showed that *V. hangzhouensis* belongs to the first subclade. In addition to the genomes of six type strains (KCTC 32373^T^, NBRC 15635^T^, 3062^T^, CGMCC 1.7062^T^, CAIM 1455^T^, and CAIM 1454^T^), seven published genomes of *Vibrio* strains designated to the Mediterranei clade were retrieved from NCBI datasets. The extended ML phylogenomic tree including JCM 19239 and JCM 19240 genomes, previously reported as two potential new species [6], clearly showed that strains CB1-14^T^ and CB2-10 retain their distinct phylogenomic position within this clade (Appendix A).

Moreover, the CB1-14^T^ genome showed values of the overall genomic relatedness indices (OGRIs) (Figure 2, Appendix A) below the species boundary of 95–96% ANI/AAI and 70% dDDH thresholds [47,48,49]: the highest ANI/AAI values were 91.3%/92.7% with *V. variabilis* CAIM 1454^T^ and 90.3%/93.1% with *V. maritimus* CAIM 1455^T^ (Figure 2B). The dDDH values (formula d4) between strain CB1-14^T^ and the Mediterranei clade members ranged from 20.9% (*V. thalassae* CECT 8203^T^) to 45.7% (*V. maritimus* CAIM 1455^T^) (Figure 2A). The OGRIs between the two new strains CB1-14^T^ and CB2-10 were 99.7%/99.2% (ANI/AAI) and 96.8% (dDDH) (Figure 2A,B).

Taken together, according to the phylogenomic position and OGRI values, the new strains CB1-14^T^ and CB2-10 are predicted to represent a novel species of the genus *Vibrio*.

### 3.2. Genomic Characteristics and Pan-Genome Analysis

The complete genomes of strains CB1-14^T^ (=KMM 8419^T^) and CB2-10 (=KMM 8420) were assembled de novo into two circular chromosomes with one plasmid for each strain (Figure 3A). Genomic features and assembly statistics are detailed in Table 1. The characteristics corresponded to the updated proposed minimal standards for genomic data usage in the current prokaryotic taxonomy [50]. The CB1-14^T^ and CB2-10 genomes are 5,591,686 bp and 5,536,255 bp in size, respectively, with an overall GC content of 46.1%. They code 4782 and 4705 proteins, respectively. The chromosomal level of genome assembly allowed accurate calculation of the number of rrn operons in each strain (Table 1); all operons were found on the leading strands and were concentrated around the origin of replication (Figure 3A).

To determine the genetic heterogeneity of the Mediterranei clade, the pan-genome of eight type strains and two new strains was performed (Figure 2B). This pan-genome, presented by gene clusters (GCs) of orthologous protein groups, comprised 9401 gene clusters with 54,917 gene calls. The core genome of the clade included 3156 gene clusters (27,666 genes found in all genomes), while the accessory genome included 1625 shell gene clusters (7399 genes) and 4620 cloud gene clusters (6100 genes). Unique genes (called singletons) belong to the cloud and are strain-specific. Analysis of the accessory genome revealed some differences both between the type strains (species-specific) and between the new isolates, supporting their strain status (Figure 3B,C). Strain-specific differences between CB1-14^T^ and CB2-10 can be defined by M00061 (D-glucuronate degradation), M00127 (thiamine biosynthesis), and M00881 (lipoic acid biosynthesis) (Figure 3C). Very recently, it has been shown that strain-specific genes of CB1-14^T^ were sulfotransferase genes and other ones related to biosynthesis of a sulfated capsular polysaccharide [29]. Furthermore, species-specific genes were related to M00015 (proline biosynthesis, absent in *V. maritimus*), M00033 (ectoine biosynthesis, present in *V. variabilis*), M00552 (D-galactonate degradation, De Ley–Doudoroff pathway, present in *V. barjaei*), and M00533 (homoprotocatechuate degradation, present in *V. hangzhouensis*).

Moreover, the core, shell, cloud, and unique gene clusters of the Mediterranei clade were annotated into COG classes (Figure 4A). Among them, the most represented functional classes (6% or more) were amino acid metabolism and transport (E), carbohydrate metabolism and transport (G), transcription (K), cell wall/membrane/envelope biogenesis (M), and general functional prediction only (R). The most abundant COG in the core genome was translation (9.5%), while in both shell and cloud gene clusters, COG carbohydrate metabolism and transport counted for 22% and 12.3%, respectively. It is worth noting that each of the *Vibrio* genomes contained from 183 to 780 unique genes (Figure 4B).

The largest number of unique genes, including paralogs, was observed in genomes of *V. thalassae* CECT 8203^T^ (780 genes) and *V. maritimus* CAIM 1455^T^ (778), followed by *V. hangzhouensis* CGMCC 1.7062^T^ (643), *V. variabilis* CAIM 1454^T^ (451), *V. mediterranei* NBRC 15635^T^ (317), and *V. barjaei* 3062^T^ (312). The genomes of CB1-14^T^ and CB2-10 accounted for the smallest number of unique genes, 204 and 183, respectively. According to the COG class annotation of these unique genes (Figure 4B), the most abundant functional classes were carbohydrate metabolism and transport, transcription, cell wall/membrane/envelope biogenesis, general functional prediction only, and mobilome: prophages.

### 3.3. In Silico Analysis of Hydrolytic and Biosynthetic Potentials 

The genomic comparison of CAZymes repertoires (CAZomes) revealed that CB1-14^T^ and CB2-10 genomes encode 130 and 127 CAZymes (Figure 5A), involved in polysaccharide degradation, transport, and regulation. They included 73 and 74 glycoside hydrolases (GHs), 37 and 34 glycoside transferases (GTs), eight polysaccharide lyases (PLs) for each strain, eight and seven carbohydrate esterases (CEs), and four auxiliary activities (AAs) for each strain, respectively.

The most abundant GHs in the *Vibrio* genomes of the Mediterranei clade members were GH13, GH23, GH3, and GH1. Strains CB1-14^T^ and CB2-10 harbored 14 GH13, which are involved in the hydrolyzing of α-linkages in glucans, indicating their potential to degrade starch and pullulan. The GH23 family was represented by six and eight putative lytic transglycosylases with an activity on peptidoglycans. Moreover, GH23 has also been found to have chitinase activity [51]. The GH3 family was represented by five endo-β-1,3-glucanases capable of laminarin degrading of marine micro- and macroalgae. Recently, GH3 isolated from a bioluminescent *V. campbellii* strain was shown to utilize the algal storage glucan laminarin [52]. Strains CB1-14^T^ and CB2-10 shared the largest number of GH94 compared to other clade members. These enzymes reversibly catalyze the phosphorolysis of β-glycosides and may take part in degradation of certain mannan oligosaccharides [53]. Additionally, CB1-14^T^ and CB2-10 differed in the presence/absence of GH24, which targets peptidoglycan, and GH109, which has acetylgalactosaminidase activity. It is interesting to note that the GH43 family containing arabinases and xylosidases was predicted only in CB1-14^T^ and CB2-10, as well as in *V. maritimus* CAIM 1455^T^ isolated from zoanthid *Palythoa caribaeorum* mucus. This indicates the potential of those strains to break down xylans and pectins.

The second class of enzymes most represented in the *Vibrio* CAZomes were GTs (Figure 5A). Among them, GT2 and GT4 accounted for the highest proportion, followed by GT51, GT83, and GT119. It has been shown that GT2 and GT4 perform the synthesis of α- and β-glycans and glycoconjugates both in Gram-positive and Gram-negative bacteria. Most of the annotated polysaccharide lyases were classified as PL7 and PL17, which are commonly found in marine bacteria and possess alginate lytic activity [54]. Previously, it was shown that many widespread *Vibrio* species could utilize alginate as a carbon source [55]. Among carbohydrate esterases, CE4 and CE9 were the most abundant. They included xylan-degrading enzymes and N-acetylglucosamine-6-phosphate deacetylases, which are important for the metabolism of chitin. Genome comparison also showed that strains CB1-14^T^ and CB2-10 contain AAs, which include chitin- and cellulose-active polysaccharide monooxygenases (AA10). Moreover, in the CB1-14^T^ and CB2-10 genomes, polysaccharide utilization locus (PUL) and several CBMs (CBM5, CBM6) coordinating the activities of enzymes targeting chitin were found (Figure 5B).

Overall, the number of CAZymes produced by CB1-14^T^ and CB2-10 is not significant, accounting for 2.5%, but predicted enzymes indicate that these strains have the potential ability to degrade alginate, xylan, and chitin, common polysaccharides in marine ecosystems. Moreover, the abundance of GHs dedicated to starch degradation demonstrated possible adaptation of *Vibrio* species toward plant polysaccharides in aquatic environments near shorelines and freshwater sources.

In silico secondary metabolite prediction analysis using the AntiSMASH tool [44] found up to five biosynthetic gene clusters (BGS) across the *Vibrio* genome sequences. Two clusters similar to betalacton and RiPP-like type were revealed in all strains except for CB1-14^T^ (Figure 5C). The cluster involved in the biosynthesis of nucleosides was detected only in the genomes of members of subclade 2 (CB1-14^T^, CB2-10, CAIM 1455^T^, and CAIM 1454^T^).

Nine putative transport secretion systems (TSS), essential in global cell functions, were detected in the *Vibrio* genomes using the MacSyFinder tool [45]. The most *Vibrio* strains shared T1SS, T2SS, T4aP, T5cSS, and the flagellar system. The maximum number of genes was found in T1SS, which is widespread in Gram-negative bacteria and mediates one-step protein translocation [56]. *V. hangzhouensis* CGMCC1.7062^T^ uniquely possessed the autotransporter T5aSS, which is essential in bacterial virulence [57]. CB1-14^T^ and CB2-10 were distinguished from other *Vibrio* strains by the absence of all components of both systems MSH (type 4 pili family) and T6SSi (cell-contact-dependent killing).

### 3.4. Phenotypic Characterization and Chemotaxonomy

The cells of the KMM 8419^T^ (=CB1-14^T^) and KMM 8420 (=CB2-10) are small, rod-shaped, 0.5 µm wide and 1–2 µm long, motile by from 2 to 14 flagella located polarly and/or laterally (Figure 6), and oxidase- and catalase-positive. Novel bacteria could grow in salinities from 0.5 to 7% NaCl and grew well on/in SWM, MA 2216, and MB 2216.

Strains KMM 8419^T^ and KMM 8420 demonstrated a positive reaction of DNA, tyrosine, Tween 80, Tween 40, Tween 20, gelatin, and starch hydrolysis, and resistance to kanamycin, carbenicillin, and oxacillin (Table 2), but differed in their ability to utilize carbon sources in API 20E/20NE/ZYM tests (Appendix A). Strains KMM 8419^T^ and KMM 8420 were similar to *V. thalassae* KCTC 32373^T^ in the requirement of NaCl for growth, the ability to hydrolyze starch, tyrosine, and DNA and to produce H_2_S, but they had distinctive carbohydrate utilization patterns compared with those of other representatives of the Mediterranei clade (Table 2).

The predominant fatty acids contained in strains KMM 8419^T^ and KMM 8420 weredetected to be C_16:1_*ω*9*c* (20.34% and 20.95%), followed by C_16:1_*ω*7*c* (17.96% and 18.89%), C_16:0_ (11.79% and 12.56%), C_18:1_*ω*9*c* (9.56% and 8.83%), C_18:1_*ω*7*c* (7.67% and 7.51%), C_14:0_ (5.67% and 5.50%), and isoC_16:0_ (5.80% and 4.61%), respectively (Table 3). The fatty acid profile of the studied strains was similar to that of *V. thalassae* KCTC 32373^T^ in its contents of C_14:0_ (5.56%), C_16:1_*ω*9*_c_* (24.89%), C_16:0_ (12.40%), but differed in the content of C_16:1_*ω*7*c* (5.67%), C_18:1_*ω*9*c* (16.70%), and C_18:1_*ω*7*c* (0.98%) (Table 3). As indicated in the original descriptions for other species of the Mediterranei clade, the strains were characterized by the abundant content of C_16:1_*ω*9*c*/C_16:1_*ω*7*c* and C_18:1_*ω*9*c*/C_18:1_*ω*7*c*, and the presence of C_14:0_ (5.7–11.8%) and C_16:0_ (10.0–21.2%) [9,10,11,12].

Strains KMM 8419^T^ and KMM 8420 included the predominant quinone Q-8, and the polar lipids were phosphatidylethanolamine (PE), lysophosphatidylethanolamine (LPE), phosphatidylglycerol (PG), diphosphatidylglycerol (DPG), two unidentified aminolipids (AL1, AL2), two unidentified phospholipids (PL1, PL2), an unidentified aminophospholipid (APL), monohexosyldiacylglycerol (MHDG), and hexuronyldiacylglycerol (HuDG) (Appendix A). Strain KMM 8419^T^ was characterized by the presence of nine unidentified lipids (L1–L9) and phosphatidic acid (PA), whereas strain KMM 8420 had eleven unidentified lipids (L1–L11) and no phosphatidic acid (PA). The type strain *V. thalassae* KCTC 32373^T^ differed from the new strains by the absence of unidentified aminolipids (AL1, AL2) and phospholipids (PL1, PL2) in its lipid composition and the presence of seven unidentified lipids (L1–L7) (Appendix A).

A DNA GC content of 46.1% was calculated from genome sequences of strains KMM 8419^T^ and KMM 8420, which is close to the values of 44.0–46.5 mol% determined for the members of the genus *Vibrio* (Table 1). The phylogenetic distinctness of strains KMM 8419^T^ and KMM 8420 was supported by phenotypic differences in the temperature and salinity ranges that provided their growth, substrate hydrolysis ability, and carbohydrate utilization patterns. Differential phenotypic and physiological characteristics are indicated in Table 2 and Appendix A. Based on the combination of phylogenomic analyses and phenotypic characteristics, it is proposed to classify strains KMM 8419^T^ and KMM 8420 as a novel species, *Vibrio chaetopteri*, with the type strain KMM 8419^T^.

## 4. Conclusions

Description of *Vibrio chaetopteri* sp. nov.:

*Vibrio chaetopteri* (chae.top’te.ri. N.L. gen. masc. n. *chaetopteri*, of *Chaetopterus*, pertaining to the source of isolation of strains).

Gram-negative, aerobic, oxidase- and catalase-positive, with rod-shaped cells, 0.5–0.8 μm in diameter and 1.2–2.0 μm in length, encapsulated, motile by from 2 to 14 polar and/or lateral flagella. On MA 2216, produces cream-pigmented opaque shining colonies. Growth occurs in 0.5–7% NaCl (optimal is 2–3%) and at 10–32 °C (optimal is 23–25 °C), no growth at 40 °C. The pH range for growth is 5–11 with an optimum of 7.0–8.0.

Negative for hydrolysis of casein and chitin; positive for DNA, starch, tyrosine, gelatin, Tween 20, Tween 40, Tween 80 hydrolysis, H_2_S production, and nitrate reduction in conventional tests.

According to the API 20E test, positive for indole production, ONPG, oxidation of D-glucose, D-mannitol, D-sucrose, amygdalin, reduction of nitrates to nitrites; weak reaction for tryptophane deaminase, L-rhamnose; and negative for arginine dihydrolase, lysine decarboxylase, ornithine decarboxylase, citrate utilization, urease production and H_2_S production under anaerobic conditions, acetoin production (Voges–Proskauer reaction), and oxidation of L-arabinose and D-sorbitol; reactions for oxidation of inositol and D-melibiose were strain-dependent (the type strain reaction was negative).

According to the API 20NE test, positive for indole production, reduction of nitrates to nitrites, hydrolysis of esculin, PNPG, and negative for arginine dihydrolase, urease production, fermentation of D-glucose under anaerobic conditions, assimilation of D-glucose, L-arabinose, D-mannose, D-mannitol, N-acetylglucosamine, D-maltose, D-gluconate, caprate, adipate, L-malate, citrate, and phenylacetate.

According to the API ZYM test, alkaline phosphatase, esterase (C 4), esterase lipase (C 8), leucine arylamidase, acid phosphatase, and naphtol-AS-BI-phosphohydrolase activities are present, but valine arylamidase and N-acetyl-β-glucosaminidase are weakly active; α-galactosidase, β-galactosidase, β-glucuronidase, cystine arylamidase, α-chymotrypsin, β-glucosidase, α-mannosidase, and α-fucosidase are not active; activities for lipase (C 14), trypsin, α-glucosidase were strain-dependent (the type strain was active for lipase (C 14), weakly positive for α-glucosidase, and not active for trypsin).

The DNA GC content of 46.1% is calculated from the genome sequence. Major fatty acids are C_16:1_*ω*9*c*, C_16:1_*ω*7*c*, and C_18:1_*ω*9*c*, followed by C_16:0_ and C_18:1_*ω*7*c*.

The GenBank accession numbers for the whole-genome sequences of strains CB1-14^T^ (=KMM 8419^T^) and CB2-10 (=KMM 8420) are CP115920.1–CP115922.1 and CP183055.1–CP183057.1, respectively.

## Figures and Tables

**Figure 1 microorganisms-13-00638-f001:**
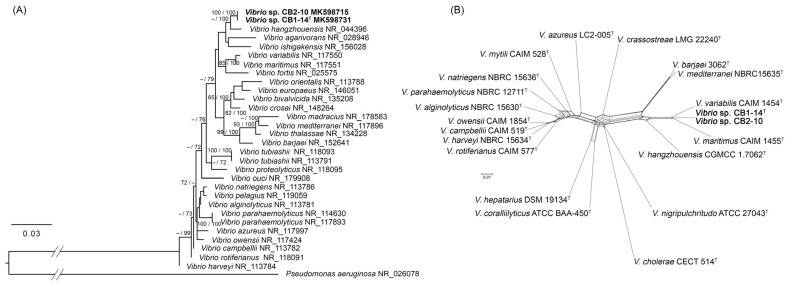
Position of new strains CB1-14^T^ (=KMM 8419^T^) and CB2-10 (=KMM 8420) and related type strains of the genus *Vibrio* based on 16S rRNA ML/MP phylogenetic tree (**A**) and five-gene MLST neighbor-net phylogenetic network (**B**). The 16S rRNA tree was inferred under the GTR + GAMMA model, and the numbers above the branches represent support values when they exceed 60% with ML (left) and MP (right) bootstrapping of 1000 replicates. Neighbor-net analysis was performed with a Jukes–Cantor correction. The bars indicate 0.03 (**A**) and 0.01 (**B**) accumulated substitutions per nucleotide position.

**Figure 2 microorganisms-13-00638-f002:**
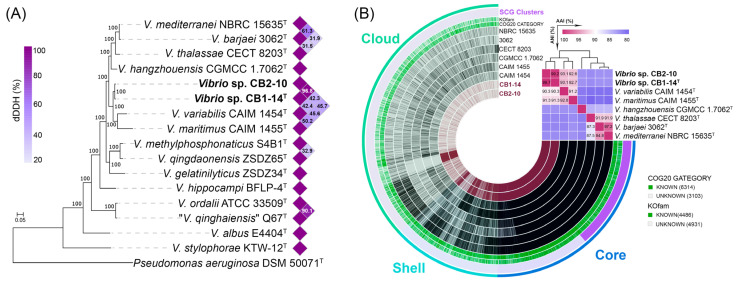
(**A**) ML genomic tree showing the phylogenetic position of new strains CB1-14^T^ and CB2-10 and related type strains of the genus *Vibrio* based on concatenated sequences of 400 translated proteins. Bootstrap values are based on 100 replicates. Bar, 0.05 substitutions per amino acid position. (**B**) The pan-genome of type strains of seven *Vibrio* spp. affiliated to the Mediterranei clade generated with Anvi’o [39]. Circle bars represent the presence/absence of 9401 pan-genomic clusters in each genome. Gene clusters are organized as core, shell, and cloud ones using Euclidian distance and Ward ordination. The heatmap in the upper right corner displays pairwise ANI and AAI values in percentages.

**Figure 3 microorganisms-13-00638-f003:**
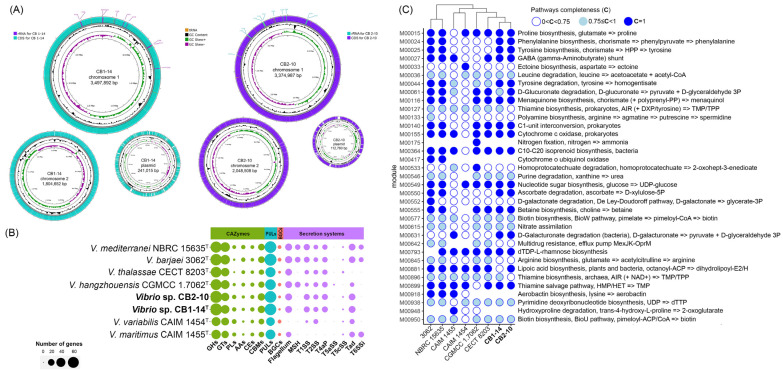
(**A**) Chromosome and plasmid maps of CB1-14^T^ (green) and CB2-10 (violet) created by Proksee server [46]. The first circle represents GC skew (in violet and green), followed by the circle of GC content (in black). *rrn* operons are shown as violet and green colors. (**B**) Gene contents and distribution of CAZymes, PULs, BGCs, and secretion systems in the Mediterranei clade pan-genome. (**C**) Discrimination of the Mediterranei clade species based on completeness of predicted KEGG pathway modules.

**Figure 4 microorganisms-13-00638-f004:**
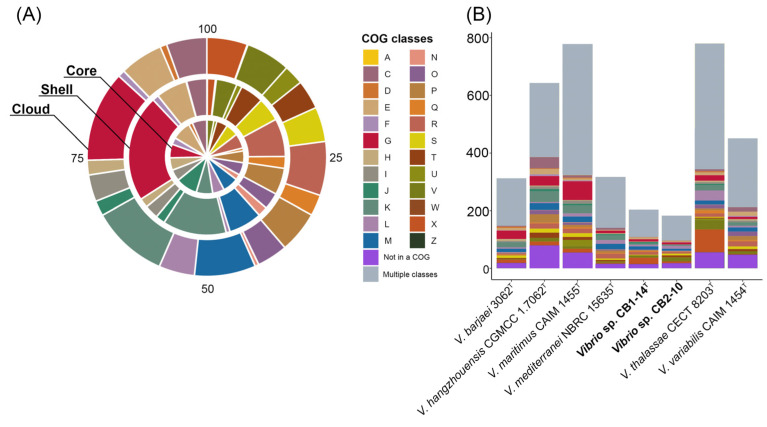
COG classes predicted in the core, shell, and cloud genomes (**A**) and number of unique genes assigned to a functional class (COG) among strains of the Mediterranei clade (**B**). Classes: A, RNA processing and modification; C, energy production and conversion; D, cell cycle control and mitosis; E, amino acid metabolism and transport; F, nucleotide metabolism and transport; G, carbohydrate metabolism and transport; H, coenzyme metabolism; I, lipid metabolism; J, translation; K, transcription; L, replication and repair; M, cell wall/membrane/envelope biogenesis; N, cell motility; O, post-translational modification, protein turnover, chaperone functions; P, inorganic ion transport and metabolism; Q, secondary structure; R, general functional prediction only; S, function unknown; T, signal transduction; U, intracellular trafficking and secretion; V, defense mechanisms; W, extracellular structures; X, mobilome: prophages, transposons; Z, cytoskeleton; Multiple classes, genes assigned to two or more COG categories; Not in a COG, COG not defined.

**Figure 5 microorganisms-13-00638-f005:**
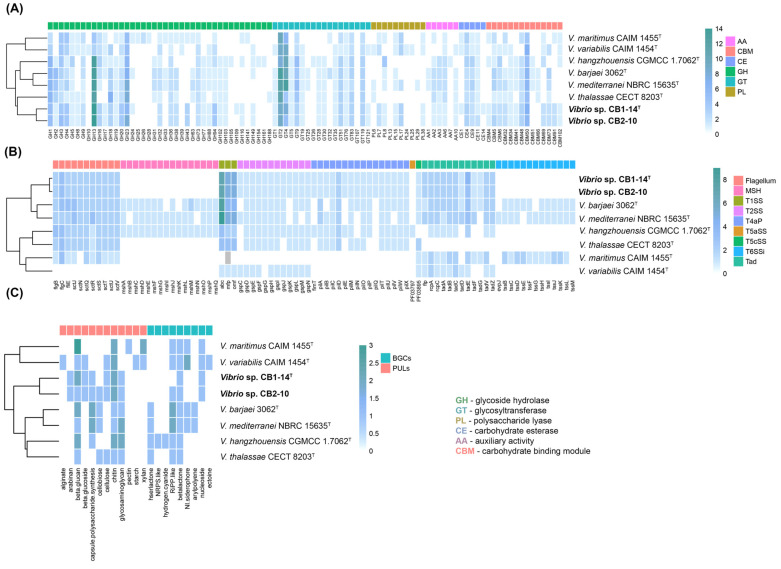
Distribution of CAZymes, PULs, and biosynthetic and secretion system gene clusters in CB1-14^T^, CB2-10, and type strains of the Mediterranei clade. (**A**) Heatmap of CAZyme family abundance in *Vibrio* species. (**B**) Heatmap of secretion system gene clusters in *Vibrio* species. (**C**) Heatmap of PULS and biosynthetic gene clusters in *Vibrio* species.

**Figure 6 microorganisms-13-00638-f006:**
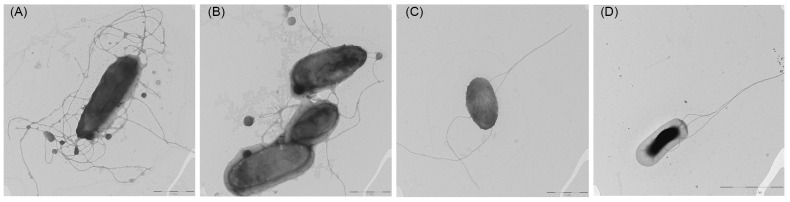
Transmission electron micrographs of strains KMM 8419^T^ (**A**–**C**, bar, 1 µm) and KMM 8420 (**D**, bar, 2 µm).

**Table 1 microorganisms-13-00638-t001:** Genomic features of CB1-14^T^, CB2-10, and type strains of the Mediterranei clade.

Feature	1	2	3	4	5	6	7	8
Assembly level	chromosome	chromosome	scaffold	contig	scaffold	contig	contig	contig
Genome size (Mb)	5.5	5.5	5.1	5.6	5.5	5.1	5.8	5.1
Number of contigs	3	3	207	67	67	60	97	158
G+C Content (mol%)	46.1	46.1	44.5	44.0	44.0	46.5	46.0	46.5
N50 (Kb)	3497.9	3375.0	74.8	241.4	286.2	171.2	207.3	81.6
L50	1	1	17	8	7	10	8	16
Coverage	141.0×	116.0×	92.0×	165×	100.0×	246.0×	31.0×	17×
Total genes	5015	5028	4728	5217	5052	4718	5523	4934
Protein coding genes	4782	4705	4635	5107	4961	4631	4151	3841
rRNAs (5S/16S/23S)	10/10/10	11/11/11	6/1/0	2/2/2	1/1/1	2/6/1	1/2/1	2/4/2
tRNAs	113	110	85	62	39	56		75
checkM completeness (%)	98.62	97.61	98.51	99.62	99.61	99.55	69.68	71.75
checkM contamination (%)	3.15	2.78	1.82	2.04	2.60	3.79	1.55	1.48
WGS project/RefSeq	GCF_040412085.2	GCF_048400125.1	OANU01	BCUE01	LQXO02	FNVG01	JYJJ01	JYJK01
Genome assembly	ASM4041208v1	ASM4840012v1	VthalassaeCECT 8203_ Velvet_Prokka	ASM159112v1	ASM163906v2	IMG-taxon 2617270906	ASM326377v1	ASM326378v1

Strains: 1, CB1-14^T^; 2, CB2-10; 3, *V. thalassae* CECT 8203^T^; 4, *V. mediterranei* NBRC 15635^T^; 5, *V. barjaei* 3062^T^; 6, *V. hangzhouensis* CGMCC 1.7062^T^; 7, *V. maritimus* CAIM 1455^T^; 8, *V. variabilis* CAIM 1454^T^.

**Table 2 microorganisms-13-00638-t002:** Differential characteristics of strains KMM 8419^T^ and KMM 8420 and type strains of the Mediterranei clade.

Feature	1	2	3	4	5	6	7	8
Growth at/in:								
10 °C	+	+	−	ND	−	−	+	+
28 °C	+	+	+	+	+	+	+	+
32 °C	+	+	+	ND	+	+	+	+
0.5% NaCl	+	+	+	ND	−	+	+	+
3% NaCl	+	+	+	+	+	+	+	+
6% NaCl	+	+	−	+	+	+	+	+
pH	5–11	5–11	5–10.5	ND	ND	6–10	5–12	6–12
Tyrosine hydrolysis	+	+	+	ND	ND	+	ND	ND
Starch hydrolysis	+	+	+	+	+	+	ND	+
Gelatin hydrolysis	+	+	−	−	+	+	ND	+
Tween 80 hydrolysis	+	+	−	+	+	−	+	+
Tween 40 hydrolysis	+	+	ND	ND	ND	ND	ND	ND
Tween 20 hydrolysis	+	+	ND	ND	ND	+	ND	ND
DNA hydrolysis	+	+	+	+	ND	−	ND	ND
H_2_S production	+	+	+	ND	−	+	−	−
Nitrate reduction	+	+	+	ND	ND	−	+	+
API 20E tests:								
ONPG	+	+	+	ND	+	ND	+	+
Citrate	−	−	−	ND	−	+	−	−
Urease	−	−	−	ND	−	−	−	−
Glucose	+	+	+	ND	+	ND	+	+
Mannitol	+	+	+	ND	+	ND	+	+
Inositol	−	w	−	+	ND	ND	+	−
Sorbitol	−	−	−	ND	+	ND	−	−
L-rhamnose	+	+	−	ND	−	ND	−	+
D-sucrose	+	+	−	+	+	+	+	+
D-melibiose	−	w	−	ND	−	ND	−	−
Amygdalin	+	+	+	ND	+	ND	+	+
L-arabinose	−	−	−	ND	−	ND	−	ND

Strains: 1, KMM 8419^T^; 2, KMM 8420; 3, *V. thalassae* KCTC 32373^T^ (data were obtained from the present study); 4, *V. mediterranei* CECT 621^T^ [8,10]; 5, *Vibrio barjaei* 3062^T^ [10]; 6, *V. hangzhouensis* CGMCC 1.7062^T^ [11]; 7, *V. maritimus* CAIM 1455^T^ [12]; 8, *V.variabilis* CAIM 1454^T^ [12]. Symbols: (+)—positive, (−)—negative, (w)—week, ND—no data.

**Table 3 microorganisms-13-00638-t003:** Cellular fatty acid composition (%) of strains KMM 8419^T^, KMM 8420, and *V. thalassae* KCTC 32373^T^.

Fatty Acid	1	2	3
C_12:0_	3.65	3.60	2.54
C_12:0_ 3-OH	2.22	1.53	1.65
iso-C_14:0_	2.58	1.76	1.50
C_14:0_	5.67	5.50	5.56
iso-C_15:0_	0.73	1.54	0.41
anteiso-C_15:0_	0.51	0.27	1.28
C_15:0_	0.47	0.63	4.33
C_14:0_ 3-OH	2.27	1.57	2.67
iso-C_16:0_	5.80	4.61	2.35
C_16:1_*ω*9*c*	20.34	20.95	24.89
C_16:1_*ω*7*c*	17.96	18.89	5.67
C_16:0_	11.79	12.56	12.40
C_17:1_*ω*8*c*	0.34	0.46	3.62
C_18:1_*ω*9*c*	9.56	8.83	16.70
C_18:1_*ω*7*c*	7.67	7.51	0.98

Strains: 1, KMM 8419^T^; 2, KMM 8420; 3, *V. thalassae* KCTC 32373^T^ (data were obtained from the present study). Fatty acids representing < 1% in all strains tested are not shown.

## Data Availability

The type strain of the species is strain CB1-14^T^ = (KMM 8432^T^ = KCTC 92790^T^), isolated from the marine polychaete *Chaetopterus cautus* from the Sea of Japan, Russia. The DDBJ/ENA/GenBank accession numbers for the 16S rRNA gene of strains CB1-14^T^ (=KMM 8419^T^) and CB2-10 (=KMM 8420) are MK598731 and MK598715, respectively, and accession numbers for the whole-genome sequences of strains CB1-14^T^ (=KMM 8419^T^) and CB2-10 (=KMM 8420) are CP115920.1–CP115922.1 and CP183055.1–CP183057.1, respectively.

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
