# Peer review of "Description and Genome-Based Analysis of Vibrio chaetopteri sp. nov., a New Species of the Mediterranei Clade Isolated from a Marine Polychaete"

_microorganisms, 2025, doi:10.3390/microorganisms13030638_

Round 1

Reviewer 1 Report

Comments and Suggestions for Authors

Comments on the manuscript with ID (microorganisms-3408044-peer-review-v1)

The manuscript is about the description and genome-based analysis of a new species, Vibrio chaetopteri sp. nov., isolated from a marine polychaete in the Sea of Japan. The manuscript is interesting; however, there are several questions the authors should reply before the manuscript considered for publication in Microorganisms. 

Q1. The method of the Antibiotic susceptibility testing should be described in detail.

Q2. Table S1 and Figure S1 should be supplied in the main text of the manuscript instead of the supplementary materials.

Q3. How the authors handle the specimens of marine polychaetes Cheatopterus cautus before conducting the bacteriological examination?

Q4. Which genes were used in the MLSA for Vibrio chaetopteri?

Author Response

Responses to Reviewer 1.

Comment: The manuscript is about the description and genome-based analysis of a new species, Vibrio chaetopteri sp. nov., isolated from a marine polychaete in the Sea of Japan. The manuscript is interesting; however, there are several questions the authors should reply before the manuscript considered for publication in Microorganisms.

Response: Thank you very much for taking the time to review our manuscript. 

Comment 1: The method of the Antibiotic susceptibility testing should be described in detail.

Response 1: We have added more details about how antibiotic susceptibility testing was performed. Please see Lines 115-117.

Comment 2: Table S1 and Figure S1 should be supplied in the main text of the manuscript instead of the supplementary materials.

Response 2: This MLST scheme was developed for the specific task of selecting strains belonging to new species (in particular, CB1-14T) from specific clades for further genome sequencing. This study does not deal with strain typing at all. Therefore, we have included the table S1 in the Supplementary. Figure S1 is an extended version of the ML tree of the type strains of the Mediterranei clade (Fig. 2) by adding all genomes of this clade from the NCBI database. This was done to ensure that our new strains were not undescribed species. Now, Figure S1 is renamed as Figure S2.

Comment 3: How the authors handle the specimens of marine polychaetes Cheatopterus cautus before conducting the bacteriological examination?

Response 3: The marine polychaete Chaetopterus cautus was removed from its tube, washed three times with sterile seawater, and then part of the food net was separated for further homogenization and isolation of marine bacteria.

Comment 4: Which genes were used in the MLSA for Vibrio chaetopteri?

Response 4: Gene sequences encoding FtsZ (cell division protein), GyrA and GyrB (DNA gyrase, subunits A and B), MreB (cell shape-determining protein), and RpoA (DNA-directed RNA polymerase, subunit alpha) were taken for the MLSA. Please see Lines 154-156.

Reviewer 2 Report

Comments and Suggestions for Authors

The manuscript deals with the description of the proposed Vibrio chaetopteri sp. nov., using MLSA and WGS approach, besides chemical and metabolic profiling.

The manuscript is well-written and compares the new isolates with other known strains.

I'd suggest the authors move most of the information (metabolic and chemical features) summarized in the "conclusions" section into a table, to make them more schematic and accessible.

Author Response

Responses to Reviewer 2.

Comment 1: The manuscript deals with the description of the proposed Vibrio chaetopteri sp. nov., using MLSA and WGS approach, besides chemical and metabolic profiling. The manuscript is well-written and compares the new isolates with other known strains.

Response 1: Thank you very much for taking the time to review our manuscript.

Comment 2: I'd suggest the authors move most of the information (metabolic and chemical features) summarized in the "conclusions" section into a table, to make them more schematic and accessible.

Response 2: Thank you for your suggestion. However, this is a taxonomic article, where the conclusion corresponds to the detailed protologue (a set of associated elements representing the first publication of a new taxon). This type of conclusion is a mandatory part and the basis for validating the publication of the taxon name.

Reviewer 3 Report

Comments and Suggestions for Authors

In this work, the authors present results toward the characterization of a novel species within the Vibrio genus. While an interesting premise, there are several points to consider.

The authors note that determination of novelty with the Vibrio genus, and location within clades, is determined through assessment of the genome either in trees or quantitatively. As such, we’ll go through the tree types first.

-Using a 16S rRNA tree in Vibrio taxonomy is not information-bearing other than to state that the isolate is likely within the genus. It is well known that neither novelty nor phylogenetic relationship can inferred through 16S rRNA sequence comparisons. While the field does still seem to like seeing a 16S rRNA tree, and many reviewers still request 16S rRNA sequences independently deposited in NCBI (the reason for which evades this reviewer as we’ve moved far beyond that level of analysis), the presence of the trees in Fig. 1 should be relegated to the Supplemental and the MLSA from the supplemental be moved into the manuscript.

-It is suspected that the MLSA tree is Fig. S1, however, the description is identical to that of the phylogenomic tree presented in Fig. 2. For an unknown reason, the authors deemed it necessary to generate new primer sets for previously-used housekeeping genes, which utterly defeats the point of scientific reproducibility. MLSA-based taxonomy in Vibrio is based on eight housekeeping genes, the lengths of which are different than those you presented – groups have been working on this exact area for over 15 years (Jiang et al. 2022 https://doi.org/10.1007/s00284-021-02725-0). Additionally, MLSA-based taxonomy requires that the breadth of genus (and family if you want to do it right) be used in the comparison to define clades. As only a small set of species were used in the suspected MLSA tree in Fig. S1, and the comparison was conducted with the wrong primers and is missing three housekeeping genes, the MLSA results presented are not scientifically meaningful or accurate. If the authors want to determine clade standing, a full MLSA must be prepared with the proper sequences from the full complement of validated and published housekeeping genes, and the sequences will be simple to extract from their complete genome. (As an aside, the writing on line 214 states that the MLSA tree is Fig. 1B but the description of this figure states that it’s a 16S rRNA tree.)

-The phylogenomic tree in Fig. 2 also lacks a substantive number of species for comparison given that the MLSA does not properly define the most-related strains (16S rRNA trees lack the resolution to define the most closely related strains in the Vibrio genus). While phylogenomic trees are interesting, this one is not informative without either a lot more strains for comparison or a much more thorough MLSA assessment.

Unfortunately, this is where the paper falls off. Without sufficient support for a resolved phylogenetic position due to the taxonomic shortcomings listed above, the rest of the manuscript lacks support for its conclusions. Your phylogenomic tree in Fig. 2 places strain CB1-14T closest to Vibrio variabilis and Vibrio maritimus but you use Vibrio thalassae for phenotypic comparisons in the Supplemental table. Why was that done? There does not appear to be any underlying logic to the results/presentation, and the components are internally contradictory.

Without a resolved taxonomic position, it is not possible to determine the most closely related species. Without such information taxonomically, it is not known which strains are relevant for comparisons of genomic indices of relatedness (ANI/AAI) to support species novelty. Additionally, it is not possible to determine which species should be compared phenotypically to facilitate a polyphasic approach. Before determining what can be done, like the comparisons presented in Fig. 3-5, determine what should be done well and use it to support the analysis.

It is also not possible to conduct further comparisons when the data is lacking – the data pertaining to strain KMM8420 are written as “XXXXXX-XXXXXX” in the paper. There are not sufficient specifics in the Methods section to facilitate reproduction of the analytical methods by another lab. This manuscript requires real, substantive revision to be complete, accurate, and fit for publication.

Author Response

Responses to Reviewer 3.

Comment 1: In this work, the authors present results toward the characterization of a novel species within the Vibrio genus. While an interesting premise, there are several points to consider. The authors note that determination of novelty with the Vibrio genus, and location within clades, is determined through assessment of the genome either in trees or quantitatively. As such, we’ll go through the tree types first.

Response 1: Thank you very much for taking the time to review our manuscript and for critical analysis of the approaches we chose, and the results obtained.

Comment 2: Using a 16S rRNA tree in Vibrio taxonomy is not information-bearing other than to state that the isolate is likely within the genus. It is well known that neither novelty nor phylogenetic relationship can inferred through 16S rRNA sequence comparisons. While the field does still seem to like seeing a 16S rRNA tree, and many reviewers still request 16S rRNA sequences independently deposited in NCBI (the reason for which evades this reviewer as we’ve moved far beyond that level of analysis), the presence of the trees in Fig. 1 should be relegated to the Supplemental and the MLSA from the supplemental be moved into the manuscript.

Response 2: We completely agree with the reviewer`s opinion that reconstructing the Vibrio 16S rRNA tree is not useful for determination of the position of a Vibrio strain. It cannot be denied that 16S rRNA genes can serve as a guide to determine the position of a strain at the genus taxonomic level. However, it cannot be used to find taxonomic neighbors for a Vibrio strain. Therefore, phylogenetic reconstructions at the level of several genes or even the genome are recommended for these purposes.

Nevertheless, the 16S sequence analysis remains an obligatory part of the description of a new taxon, today. Moreover, to prove the authenticity of genomic data, it is necessary to compare full-length 16S rRNA sequences obtained by Sanger sequencing and NGS methods. This requirement is retained in the updated Proposed Minimal Standards for Prokaryotic Taxonomy (Riesco R, Trujillo ME., 2024).

It is worth noting that both trees (16S rRNA and MLSA) are shown in Figure 1.

Comment 3: It is suspected that the MLSA tree is Fig. S1, however, the description is identical to that of the phylogenomic tree presented in Fig. 2. For an unknown reason, the authors deemed it necessary to generate new primer sets for previously-used housekeeping genes, which utterly defeats the point of scientific reproducibility. MLSA-based taxonomy in Vibrio is based on eight housekeeping genes, the lengths of which are different than those you presented – groups have been working on this exact area for over 15 years (Jiang et al. 2022 https://doi.org/10.1007/s00284-021-02725-0). Additionally, MLSA-based taxonomy requires that the breadth of genus (and family if you want to do it right) be used in the comparison to define clades. As only a small set of species were used in the suspected MLSA tree in Fig. S1, and the comparison was conducted with the wrong primers and is missing three housekeeping genes, the MLSA results presented are not scientifically meaningful or accurate. If the authors want to determine clade standing, a full MLSA must be prepared with the proper sequences from the full complement of validated and published housekeeping genes, and the sequences will be simple to extract from their complete genome. (As an aside, the writing on line 214 states that the MLSA tree is Fig. 1B but the description of this figure states that it’s a 16S rRNA tree.).

Response 3: There was no mistake. A five-gene MLSA tree, called MLST neighbor-net phylogenetic network, was shown in Figure 1 B. The ML tree in Figure S1 represented an extended version of the ML tree (400 translated proteins) with type strains of the clade Mediterranei (Figure 2) by including all genomes of that clade from NCBI database (correctly or incorrectly identified, or with taxonomically uncertain status). This was to ensure that our new strains are not members of undescribed species. So, from this tree “strains CB1-14T and CB2-10 retain their distinct phylogenomic position”.

When developing our strain-typing scheme, we were guided by the following points: the size of an amplified fragment should not exceed a Sanger sequencing run (750-800 bp), satisfactory PCR fragment production and the smallest number of genes possible. Yes, the typing scheme (Jiang, 2013) was indeed used as a basis. A lot of work was done, as a result, four genes were selected and another one was added (gyrA), which had not previously been used in any Vibrio typing scheme. It is worth noting, that four- or five-gene MLST schemes were used in many early works to study Vibrio clades (Cano-Gomez, 2011, Tarazona, 2015, Pérez-Cataluña, 2016, Dubert, 2016).

This is not a universal scheme; it was made for specific tasks. The purpose of the own MLST scheme was to select strains of new species (in particular, CB1-14T) from specific clades for further genome sequencing.

Here is Figure 1 caption with underlined parts related to MLST.

Figure 1. Position of new strains CB1-14T (=КММ 8419T) and CB2-10 (=KMM 8420), and related type strains of the genus Vibrio based on 16S rRNA ML/MP phylogenetic tree (A) and MLST neighbor-net phylogenetic network (B). The 16S rRNA tree was inferred under the GTR+GAMMA model and the numbers above the branches represent support values when they exceed 60% with ML (left) and MP (right) bootstrapping of 1000 replicates. Neighbor-net analysis was performed with a Jukes–Cantor correction. The bars indicate 0.03 (A) and 0.01 (B) accumulated substitutions per nucleotide position.

Comment 4: The phylogenomic tree in Fig. 2 also lacks a substantive number of species for comparison given that the MLSA does not properly define the most-related strains (16S rRNA trees lack the resolution to define the most closely related strains in the Vibrio genus). While phylogenomic trees are interesting, this one is not informative without either a lot more strains for comparison or a much more thorough MLSA assessment.

Response 4: Thanks for your comment. We performed MLSA with eight genes according to Jiang et al. (2022), covering 35 type strains of Vibrio, representing known Vibrio clades. Description of the results is placed in the corresponding part of Results (Lines 222-227). The MLST tree is placed in the Supplementary as Figure S1.

Comment 5: Unfortunately, this is where the paper falls off. Without sufficient support for a resolved phylogenetic position due to the taxonomic shortcomings listed above, the rest of the manuscript lacks support for its conclusions. Your phylogenomic tree in Fig. 2 places strain CB1-14T closest to Vibrio variabilis and Vibrio maritimus but you use Vibrio thalassae for phenotypic comparisons in the Supplemental table. Why was that done? There does not appear to be any underlying logic to the results/presentation, and the components are internally contradictory.

Response 5: Unfortunately, the reviewer was unable to understand the phylogenetic analysis provided, which consistently proves, starting with the 16S rRNA gene tree and ending with the construction of the genomic tree that the new strains belong to the clade Mediterranei, are within the clade, and represent a new Vibrio species. On the ML trees (Figure 2A and now Figure S2), we can accept all branches since they have 100% bootstap support. Vibrio variabilis and Vibrio maritimus are the closest species to CB1-14. These species are deposited in only two collections. Attempts to deliver them alive were unsuccessful. Over the past five years, we have been able to acquire only one type strain Vibrio thalassae from this clade.

From the literature, most of the tests performed in this study (Table S3) were not performed for other type strains of this clade. Therefore, we were forced to use Vibrio thalassae for these phenotypic comparisons.

Comment 6: Without a resolved taxonomic position, it is not possible to determine the most closely related species. Without such information taxonomically, it is not known which strains are relevant for comparisons of genomic indices of relatedness (ANI/AAI) to support species novelty. Additionally, it is not possible to determine which species should be compared phenotypically to facilitate a polyphasic approach. Before determining what can be done, like the comparisons presented in Fig. 3-5, determine what should be done well and use it to support the analysis.

Response 6: We strongly disagree with this comment. The phylogenetic position of the two strains was determined correctly, based on three sequential phylogenetic reconstructions. Please, see our explanation above. The obtained OGRIs also support species novelty of two new strains. Please, see Table S2.

Comment 7: It is also not possible to conduct further comparisons when the data is lacking – the data pertaining to strain KMM8420 are written as “XXXXXX-XXXXXX” in the paper. There are not sufficient specifics in the Methods section to facilitate reproduction of the analytical methods by another lab. This manuscript requires real, substantive revision to be complete, accurate, and fit for publication.

Response 7: We are still waiting for the accession number for strain KMM 8420 from the NCBI staff. We can share the data using FigShare.
